# Treatment of Human HeLa Cells with Black Raspberry Extracts Enhances the Removal of DNA Lesions by the Nucleotide Excision Repair Mechanism

**DOI:** 10.3390/antiox11112110

**Published:** 2022-10-26

**Authors:** Ana H. Sales, Marina Kolbanovskiy, Nicholas E. Geacintov, Kun-Ming Chen, Yuan-Wan Sun, Karam El-Bayoumy

**Affiliations:** 1Department of Chemistry, New York University, New York, NY 10003, USA; 2Department of Biochemistry and Molecular Biology, College of Medicine, Pennsylvania State University, Hershey, PA 17033, USA

**Keywords:** black raspberry extracts, oxidative DNA damage and repair, bulky benzo[*a*]pyrene DNA adducts, expression of nucleotide excision repair proteins, western blots

## Abstract

As demonstrated by us earlier and by other researchers, a diet containing freeze-dried black raspberries (BRB) inhibits DNA damage and carcinogenesis in animal models. We tested the hypothesis that the inhibition of DNA damage by BRB is due, in part, to the enhancement of DNA repair capacity evaluated in the human HeLa cell extract system, an established in vitro system for the assessment of cellular DNA repair activity. The pre-treatment of intact HeLa cells with BRB extracts (BRBE) enhances the nucleotide excision repair (NER) of a bulky deoxyguanosine adduct derived from the polycyclic aromatic carcinogen benzo[*a*]pyrene (BP-dG) by ~24%. The NER activity of an oxidatively-derived non-bulky DNA lesion, guanidinohydantoin (Gh), is also enhanced by ~24%, while its base excision repair activity is enhanced by only ~6%. Western Blot experiments indicate that the expression of selected, NER factors is also increased by BRBE treatment by ~73% (XPA), ~55% (XPB), while its effects on XPD was modest (<14%). These results demonstrate that BRBE significantly enhances the NER yields of a bulky and a non-bulky DNA lesion, and that this effect is correlated with an enhancement of expression of the critically important NER factor XPA and the helicase XPB, but not the helicase XPD.

## 1. Introduction

Human populations worldwide are exposed to a variety of naturally occurring and synthetic chemicals that include the ubiquitous environmental polycyclic aromatic hydrocarbon (PAH) pollutants [1]. PAH contaminants are also found in cigarette smoke and are known to induce cancers in preclinical animal models and contribute to the formation of human cancers as well [2]. PAHs are considered pro-carcinogens because they require metabolic activation to reactive electrophiles to exert their deleterious effects [3]. Benzo[*a*]pyrene (BP) is the most widely studied PAH and was classified by the International Agency for Research on Cancer as Group 1 (known human carcinogen [4]. Following metabolic activation of BP to diol epoxide derivatives, these intermediates react with the exocyclic amino groups of guanine to form covalent and stable DNA adducts (BP-dG) in human cells [5]. If not removed by cellular DNA repair mechanisms, the error-prone insertion of nucleotides opposite these deoxyguanosine adducts by replication polymerases, is known to lead to mutagenesis and carcinogenesis in multiple targets in preclinical animal models. BP has been used as a model carcinogen to test a variety of potentially promising cancer chemopreventive agents [2], but the specific mechanisms of chemoprevention remain unknown. 

The development of effective, safe, and easy to administer chemopreventive agents is urgently needed, and continues to be an important goal in the arena of cancer prevention research. Several sources of phytochemicals have been proposed [6], and among these candidates are black raspberries (BRB) which have shown great promise in cancer prevention in both preclinical and clinical investigations [2,7,8].

BRB is also known to inhibit Phase I enzymes and to enhance Phase II detoxification enzymes that are involved in the metabolism of polycyclic aromatic hydrocarbons such as BP, which can account for the inhibitory effect of DNA adducts in vitro and in vivo [2,9] In the current study, we aimed to determine whether BRB affects the repair of carcinogen-induced DNA damage, an effective and additional prevention mechanism beyond its role in carcinogen metabolism [2]. We hypothesize that the suppression of DNA damage by BRB is, at least in part, due to an enhancement of cellular DNA repair capacity. We examined the effects of BRB extracts (BRBE) on nucleotide excision repair (NER) and base excision repair (BER) on the excision of two kinds of DNA lesions in HeLa cell extracts, an established model system for studying DNA repair activity. One of these two DNA lesions, depicted in Figure 1, is one of the known stereoisomeric BP-dG DNA adducts that is characterized by *cis* stereochemistry [10] and is a good substrate of the NER mechanism [11,12]. This adduct assumes a base-displaced intercalative conformation in double-stranded DNA [13], and was selected for these studies rather than the stereoisomeric *trans*-adduct because it is a five-fold better NER substrate [14]. 

Besides bulky covalent DNA adducts, oxidative stress [15] generates reactive oxygen species that stimulate inflammation and DNA damage [16]. The second DNA lesion employed in this study, the non-bulky DNA lesion guanidinohydantoin [17] (Gh), Figure 1, is derived from the oxidation of the mutagenic DNA lesion 8-oxo-7,8-dihydroguanine (8-oxo-dG) which is ubiquitous in mammalian cells [18]. Unlike other non-bulky DNA lesions like 8-oxo-dG that are substrates of BER only, Gh is a substrate of human BER and NER mechanisms [19]. Therefore, Gh is of interest because it offers an opportunity to determine whether BRBE treatment affects both major DNA repair mechanisms, NER and BER. Western Blot experiments were conducted to determine whether BRBE treatment affects the expression of several critical protein factors that contribute to successful NER activity. 

### 1.1. BRBE Preparation and Dosage

BRBE was prepared as described previously [20,21]. Briefly, freeze-dried BRB powder was purchased from Decker Farms, Inc. and Berri Products, LLC. For the preparation of BRBE, BRB powder was dissolved in 80% ethanol-H_2_O, filtered by passage through a MILLEX-GP 0.22uM filter (Millipore), and used at a concentration of 160 μg/mL in culture media for 24 h; we emphasize that this dose was selected based on our extensive published studies that showed significant dosage-dependent inhibition of carcinogen-induced DNA damage and oxidative stress at BRBE dosages up to 160 μg/mL that leveled off at higher concentrations [2,21,22].

### 1.2. DNA Substrates

The oligonucleotides containing the BP-derived DNA adducts were obtained by reacting (+)-*cis*-7*R,8S*-epoxy*,9S,9R*-dihydrodiol-BP with guanine in the 11-mer oligonucleotide sequence 5′-CCATCGCTACC as described in detail elsewhere [23]. The integrity of the ligated 147-mer sequences was verified by denaturing gel electrophoresis methods and yielded single bands, as demonstrated earlier in analogous control experiments [19]. The B-dG adducts used in this work were characterized by (+)-*cis*-stereochemistry with a base-displaced intercalative adduct conformation [13]. The Gh lesions were obtained by selectively oxidizing the 8-oxo-dG residue in the oligonucleotide sequence 5′-CCATC [8-oxo-dG]CTACC as described in the literature [24]. The 11-mer sequences were 5′-endlabeled and incorporated into 147-mer duplexes by ligation method as described [25]. The radioactively labeled 11-mer sequences were ligated to the 5′-flanking 60-mer 5′-CACAGGATGTATATATCTGACACGTGCCTGGAGACTAGGGAGTAATCCCCTTG and GCGGTTACTTGGCGGTTA sequence, and to the 76-mer 5′-ACAGCGCGTACGTGCGTTTAAGCGGTGCTAGAG and CTGTCTACGACCAATTGAGCGGCCTCGGCACCGG GATTCTCCA, on its 3′-side. The 147-mers containing the lesions were subsequently purified by denaturing polyacrylamide gel electrophoresis methods and annealed to a fully complementary 147-mer strand that included C opposite the modified guanine residues. The 147-mer duplexes containing single Gh lesions or BP-dG adducts were constructed and characterized as described earlier [19].

### 1.3. Cell Culture and BRBE Treatment

This cell culture model has been established and successfully used to study the repair capacity of NER and BER enzymes in vitro [12,26]. HeLa S3 cells (ATCC CCL-2.2) were grown in F-12K medium (ATCC) supplemented with 10% fetal bovine serum (Gibco) and maintained in an atmosphere of 5% CO_2_ at 37 °C. In our experiments, ~15 × 10^6^ cell were seeded in T500 Nunclon flasks (Thermo Scientific, Waltham, MA, USA). After incubation, the BRBE solution was first filtered by passage through a MILLEX-GP 0.22 μM filter (Millipore), and an appropriate aliquot was added to the HeLa cell suspensions to generate final concentrations of 160 μg/mL. After a 24 h incubation period, the cells were harvested by the addition of a TrypLE * EXPRESS solution (Gibco) to cover all three-layer surfaces of the T500 Nunclon flasks, and resuspended in 40 mL of medium, and finally pelleted by centrifugation (1000× *g*, 10 min, 4 °C). The pellets were rinsed with 10 mL cold PBS and transferred to a 15 mL tube and pelleted at 1000× *g* (10 min, 4 °C). The pellets were snap-frozen in liquid nitrogen and stored at −80 °C.

### 1.4. Mini-Whole Cell Extract Preparation and NER Excision Product Assays

The methods of Smeaton et al. [26] were adopted to monitor the levels of the characteristic NER dual incision products. The HeLa S3 cell pellets (0.450 mL) were resuspended in hypotonic lysis buffer (10 mM Tris–HCl pH 8.0, 1 mM EDTA, 5 mM DTT), and supplemented with a protease inhibitor cocktail (Roche, San Francisco, CA, USA). A sucrose–glycerol buffer (50 mM Tris–HCl pH 8, 10 mM MgCl_2_, 2 mM DTT, 25% sucrose (*w*/*v*), 50% glycerol (*v*/*v*)) was slowly added. After complete homogenization by pipetting, a saturated (NH_4_)_2_SO_4_ solution (pH 7.0) was added slowly and mixed by rotation for 30 min, at 4 °C. The insoluble material was removed by microcentrifugation (20 min, 21,000× *g*, 4 °C) and the supernatant was transferred to a fresh tube. A saturated solution (NH_4_)_2_SO_4_, pH 7.0 solution was then added slowly (1:1 supernatant volume) and mixed by rotation for 30 min at 4 °C. After microcentrifugation for 20 min at 21,000× *g*, the proteins were collected by microcentrifugation (20 min, 21,000× *g*, 4 °C) and re-suspended in 0.4 mL of NER dialysis buffer (25 mM HEPES, pH 7.9, 100 mM KCl, 12 mM MgCl_2_, 0.5 mM EDTA, 2 mM DTT, 12% glycerol) and dialyzed against 200 mL of NER dialysis buffer (overnight, 4 °C) using the Slide-A-Lyzer dialysis cassette G2 (10,000 Molecular Weight Cut-off, Thermo Scientific).

The NER excision product assays were performed with 20 fmol of the ^32^P-labeled double-stranded DNA molecules containing the DNA lesions in the radioactively labeled strand, dissolved in 100 μL volumes of solutions (25 mM HEPES, pH 7.9, 70 mM KCl, 4 mM MgCl2, 0.1 mM EDTA, 1 mM DTT, 4 mM ATP, 200 μg/μL bovine serum albumin, 4% glycerol, and 30 μL of HeLa cell extract (12 mg/mL concentration). The reactions were terminated after pre-determined lengths of incubation times by the addition of 40 μg of proteinase K in 0.3% SDS incubated for 30 min at 37 °C. Following a phenol/chloroform extraction, the DNA incision products were ethanol-precipitated and, after re-dissolving the DNA, the solutions were subjected to analysis by denaturing 12% polyacrylamide gel electrophoresis. The polyacrylamide gels were dried and exposed to *Molecular Dynamics* Storage Phosphor screens and the images were quantitatively analyzed using a Typhoon FLA 9000 laser scanner. The gel autoradiographs were subsequently analyzed quantitatively by standard densitometry methods to determine the fractions of incision products. The NER products were easily distinguishable from BER products because of the different sizes of the excision products.

### 1.5. Western Blots

The impact of BRBE treatment of HeLa cells on the expression of the NER proteins XPB, XPA and XPD was monitored by Western Blotting. The Laemmli sample buffer (Biorad #1610737) was added to the samples of BRBE-treated or untreated HeLa cell extracts and heated for 5 min to 95 °C. The proteins, contained in 50 μg aliquots, were separated from one another by SDS-PAGE methods and the samples were transferred to PVDF membranes (Amersham^TM^ Hybond^TM^). The blots were stained with XPA (sc-28353), XPB (sc-271500), *anti*-XPD (sc-101174), and *anti*-Tubulin (sc-53140) primary monoclonal antibodies (Santa Cruz Biotechnology, Santa Cruz, CA, USA). The membranes were incubated at 4 °C overnight, and subsequently rinsed with a TBST solution (20 mM Tris-HCl, 137 mM NaCl, and 0.1% Tween 20), and incubated with the secondary antibody (sc-516102) conjugated to Horseradish Peroxidase (HRP). The signal was detected by enhanced chemiluminescence (ECL, Thermo Scientific SuperSignal West Pico PLUS). The Chemiluminescence blots were scanned using a C-DiGit Blot Scanner (LI-COR Biosciences, Lincoln, NE, USA) and analyzed by the Image Studio version 5.2 software.

## 2. Results

We investigated the effects of BRBE treatment of human HeLa S3 cells on their capacities to excise single BP-dG or Gh lesions embedded in otherwise identical 147-mer duplexes by NER and/or BER mechanisms. The lesions were positioned at the 66th nucleotide counted from the 5′-end that was ^32^P-end-labeled. The effects of pre-treatment of human cells with BRBE were investigated by exposing the HeLa S3 cells to BRBE for 24 h. The NER/BER capacities of the cells were then evaluated by incubating the modified 147-mer DNA duplexes in protein extracts obtained from BRBE-treated or untreated HeLa cells. The NER activity associated with oligonucleotides containing BP-dG, Gh, or other NER substrates, typically yield short oligonucleotide excision products ~24–30 nucleotides (nt) in length that are the hallmark of successful NER excision activity [27]. After incubation for varying time intervals, the NER excision products were recovered and visualized by gel electrophoresis imaging methods [26]. The subsequent steps of the full repair mechanism that include the re-synthesis of the gap created by the excision of the damaged oligonucleotides, have not been assessed here because the recognition and excision of the lesions are the key intermediate steps in NER and BER that were of primary interest. Finally, identical, but unmodified oligonucleotides (without lesions) are not incised, and remain intact after incubation in the same cell extracts under identical conditions [28].

### 2.1. Effects of BRBE Pretreatment of HeLa Cells on the Repair of Oxidatively Generated Gh DNA Lesions

The Gh lesion is a substrate of BER and NER repair pathways [19]. A typical example of DNA repair assays, depicting the NER and BER assays of 147-mer duplexes containing a single Gh lesion is shown in Figure 2. The un-incised 147-mer strands account for the dark bands in the upper part of the gel. After incubation of the 147-mer DNA duplexes containing single Gh lesions in HeLa cell extracts, prominent excision products are visible between the 65 and 70 nt marker regions that are due to base excision repair activity as shown elsewhere [19]. The molecular weight markers (M) are depicted in lanes 1 and 10 in Figure 2. The ladder of weaker bands due to NER excision products is observed in the 24–30 nt region of the gel, as expected [19] and is denoted by a vertical bar below the 30 nt marker in Figure 2.

The densitometry traces of the gel shown in Figure 2 are depicted in Figure 3 (30 min time point). The black traces were obtained with the control untreated sample, while the red traces denote the results obtained with the BRBE-treated sample. In panel A, the densitometry trace clearly displays a sharp maximum between the un-incised DNA and NER excision products which is attributed to the BER excision product as described previously [19].

Before conducting the DNA repair assays, the optical absorbances of the BRBE-treated and -untreated cell extracts were matched in order to achieve similar protein concentrations in both cases; indeed, the red and black lines in Figure 3 almost overlap with one another, as expected. Within experimental error, there is no discernable difference in BER activities between the +BRBE and −BRBE samples (Figure 3A), thus indicating that base excision repair activity is not affected by BRBE treatment.

An amplified densitometry plot of Gh NER excision products ~24–30 nt in length is shown in Figure 3B. The levels of NER excision products are clearly higher in the case of extracts derived from the BRBE-treated than untreated HeLa cells. This difference in product yields is reproducible, as discussed in more detail below. As shown previously, unmodified DNA sequences are not incised in these NER assays [12].

### 2.2. Effects of BRBE Pretreatment of Human Cells on the NER of Bulky BP-dG Adducts

Experiments analogous to those depicted in Figure 2, but with BP-dG adducts embedded in 147-mer duplexes, are shown in the gel image in Figure 4. As expected since the BP-dG adducts are not substrates of the BER mechanism [19], the BER excision band is absent in Figure 4. The weak bands observed at about 65 nucleotides (nt) in length, are due to the degradation of the modified strand at the site of the adduct that occurs during the denaturing gel electrophoresis step. The NER dual incision oligonucleotide products 24–30 nt in length are represented by the ladder of oligonucleotide excision products shown in the lower part of the gel (Figure 4). Densitometry traces of the 30 min time point are compared with and without pre-treatment with BRBE (Figure 5).

### 2.3. Quantitative Comparisons of DNA Repair Yield Averages

Quantitative comparisons of NER and BER activities obtained in extracts from BRBE-treated and untreated HeLa cells were obtained by analysis of the densitometry tracings derived from multiple repetitive experiments. The relative yields of NER products were estimated by integrating the NER signal characterized by the appearance of lesion-containing oligonucleotides ~24–30 nt in length (e.g., Figure 3B and Figure 5B); and dividing this signal by the full integration of the densitometer traces that include the un-incised 147-mer signals. The BER yields were determined by employing the same method. The product yield ratio defined as [Y(+BRBE)/Y(−BRBE)] > 1.0 denotes the enhancement of DNA repair capacity of the BRBE-treated HeLa cells.

The effect of BRBE pre-treatment on NER yields of BP-dG adducts as a function of incubation time (10–60 min) is displayed in Figure 6A. Each data point represents the average of 28 data points (seven independent experiments with four trials each). The kinetics of NER product formation as a function of incubation time are linear at least up to 40 min [31,32]. The differences between the time-dependent product ratios within the 10–60-min incubation time interval are small, and these ratios were thus combined to yield an overall relative NER ratio [Y(+BRBE)/Y(−BRBE)]_BP_ = 1.24 ± 0.09 (two-tailed *t*-test, *p* < 0.05, 95% confidence level, Figure 6A).

The time dependence (10–60 min) of the Gh ratio [Y(+BRBE)/Y(−BRBE)]_Gh_ averages, evaluated at different time points, are summarized in Figure 6B. In the case of Gh, each individual data point in this figure represents the average and standard deviations of eight independent experiments, with 4 trials in each. The average value of the NER ratios, based on 32 experimental data points, is [(Y(+BRBE)/Y(−BRBE)]_Gh(NER)_ = 1.24 ± 0.13 (two-tailed *t*-test, *p* < 0.05, 95% confidence level). Interestingly, these mean enhancement ratios in the average NER yields are the same (~24%) as the enhancement observed in the case of the BP-dG adducts.

The analogous BER ratio associated with Gh lesions is significantly smaller since [Y(+BRBE)/Y(−BRBE)]_Gh(BER)_ = 1.06 ± 0.01 (*p*-value < 0.05) (Figure 6C).

### 2.4. Impact of BRBE Treatment on the Expression of the NER Factors XPA, XPB and XPD

A potential reason for the observed enhancement of NER lesion excision activity associated with the pre-treatment of human cells with BRBE is an enhancement of the expression of NER proteins. The DNA lesions are initially recognized by the factor XPC-Rad23B that binds to the site of the lesion. The 10-protein complex TFIIH is subsequently recruited to this complex which results in the displacement the XPC-Rad23B protein dimer [30]. The TFIIH complex contains the helicases XPB and XPD that unwind the lesion-containing double-stranded DNA region by a cooperative mechanism. The protein XPA plays a structural role in the assembly of this multi-protein-DNA complex [33] that, in turn, recruits the endonucleases XPF and XPG that catalyze the dual incisions of these oligonucleotide sequences that is followed by the excision of oligonucleotides ~24–32 nucleotides in length. We employed standard Western Blot methods for comparing the expressions of three of the critically important TFIIH proteins XPA, XPB and XPD in BRBE-treated and untreated HeLa cells; XPC-Rad23B was omitted because the binding affinity of XPC-Rad23B to structurally distorted DNA sequences is not proportional to the overall observed NER dual incision efficiencies [34], and because of its affinity for binding to unmodified DNA and to structural DNA distortions even in the absence of DNA lesions or adducts [34,35,36].

An example of Western Blot experiments using four pairs of treated and untreated samples is depicted in Figure 7. Inert tubulin protein samples were employed as controls of overall cellular protein concentrations in treated and untreated cells to ensure reproducibility (Figure 8). The protein yields in BRBE-treated and untreated control HeLa cells are summarized in Figure 9.

Compared with the untreated HeLa cells, the XPA and XPB protein expressions are remarkably up regulated in the BRBE treated group relative to the untreated HeLa cells by 72 ± 9%, while the XPB protein levels were 55 ± 4% higher. By contrast, the levels of XPD were higher in BRBE-treated cells by only 14 ± 9% which is close to the reproducibility of the (−)BRBE control experiments. Overall, these results suggest that the significant upregulation of the NER factors XPA and XPB is correlated with the NER response to BRBE treatment, while enhancement of XPD expression is close to the baseline (Figure 7). These findings clearly point to an impact of BRBE on the expressions of selected NER proteins that suggest further in-depth studies of these phenomena that were beyond the scope of this study. It should also be noted that BRBE protects cellular proteins against glycation [37].

## 3. Discussion

### 3.1. BRBE and Cancer Prevention

Numerous resources of phytochemicals have been proposed for cancer prevention [2,38]. There is strong evidence supporting the notion that the effects of BRB as antioxidants and as inhibitors of Phase I and inducers of Phase II enzymes, may, in part, account for its inhibitory impact on DNA damage, mutagenesis and carcinogenesis [2]. Nevertheless, there are few previous studies of the effects of BRB on DNA repair [39,40].

In the present report we used an established model system to study DNA repair and show that the pre-treatment of HeLa cells with BRBE exhibits a measurable effect on the repair capacity of two structurally very different NER substrates. The dose of BRBE was selected based on multiple previous studies demonstrating that 160 μg/mL was an optimal dose for the inhibition of carcinogen-induced DNA damage and oxidative stress [2,21,22]. Bulky DNA adducts generated from tobacco and environmental pollutants such as benzo[*a*]pyrene, are known to cause mutations that often drive malignant transformation.

### 3.2. The BP-dG and Gh Lesions as Model Systems

BP is a product of combustion of fossil fuels and cigarette smoke and is a ubiquitous environmental pollutant and chemical carcinogen in urban and industrial environments [41]. It is metabolized to highly reactive diol epoxide derivatives (BPDE) that bind mostly to the exocyclic amino group of guanines in DNA to form covalent DNA adducts (Figure 1). These bulky PAH-DNA adducts are substrates of the human NER system that counteracts the impact of these carcinogens on human health. Besides the modulating effects of BRBE on metabolizing enzymes that are involved in the metabolic activation and thus inhibition of DNA damage, we show here that it also enhances the NER activity that removes BP-dG and Gh lesions from DNA. In our previous study we demonstrated that BRBE inhibits the formation of bulky lesions derived from chemical (PAHs) and physical (UV) carcinogens, even when added after carcinogen treatment; based on these results, we hypothesized that BRBE could enhance the NER capacity for repairing DNA lesions [21] and inhibit mutagenesis induced by chemical and physical agents in Human Oral Leukoplakia and Rat Oral Fibroblast.

We reasoned that Gh would be interesting to study because the BER and NER mechanisms appear to compete for excising the same lesion [24,34]. In our experiments, the BER activity of Gh in our DNA repair pathways are robust (Figure 3A) and overshadows the NER activity. Thus, competition between the two repair pathways cannot account for the lack of a BRBE effect on base excision repair. We conclude that the Gh. 

BER mechanism, unlike the NER mechanism, is not sensitive to BRBE treatment within our experimental error bars (Figure 3C).

The smaller effects of the Gh BER yields (<6%) re-enforces the validity of the NER results and eliminates the possibility of a non-specific, or spurious experimental artefact that could have explained the observed enhancement in the NER response of the Gh or BP-dG lesions.

### 3.3. Impact of BRBE on the Expression of NER Proteins

Our results suggest that BRBE might affect one of the ~11 proteins that are necessary for excising the lesion-containing DNA sequences 24–30 nt in length by the NER mechanism [30,42].

The NER mechanism that leads to the excision of these characteristic NER products involves ~11 proteins (30 proteins for fully regenerating the original, undamaged DNA sequences [30]). The excision process includes the recognition of the DNA lesions by the heterodimeric factor XPC-RAD23B, the subsequent recruitment of the 10-factor protein complex TFIIH, and the eviction of XPC-RAD23B from the TFIIH-DNA lesion complex. The XPB helicase in TFIIH unwinds a small region of the DNA duplex around the site of the lesion that is followed by the ATP-powered further unwinding and enlargement of the unwound duplex region catalyzed by the helicase XPD. The stalling of XPD at the site of the lesion is a signal for the recruitment of endonucleases and other factors that excise the lesion-bearing oligonucleotides 24–30 nt in length. A third important NER factor is XPA, which is a critical scaffolding protein that plays a key role in the assembly of the proteins that are required for the incision and excision pathways. Based on this well characterized NER lesion-excision mechanisms, we selected the helicases XPB, XPD and the scaffolding protein XPA for our initial Western Blot studies to determine whether the expression of these key proteins is affected by BRBE treatment of intact HeLa cells. The observed ~72% and ~58% enhancements in the expressions of XPB and XPA, respectively, are clearly significantly higher than the 3–9% standard deviations range that characterize the reproducibility of these experiments (Figure 7). These enhancements in protein yields are not observed in the case of tubulin that was used as a negative control.

Based on these findings, we conclude that enhanced expression of the NER factors XPB and XPD may play a role in the enhancement of NER activities in BRBE-treated cells.

Thus, BRB extracts increase the overall NER activity by increasing the abundance of NER factors, and not by directly modulating the repair activity. Overall, more extensive studies in these directions that were beyond the scope of this contribution, are warranted.

## 4. Perspectives

Previous studies on the role of BRBE and its extract on DNA repair are limited [39,40] It was shown that ferulic acid (a phenolic constituent in berries), when administered to mice after a whole body gamma radiation, resulted in the disappearance of DNA strand breaks at a faster rate than in unirradiated controls, suggesting enhanced DNA strand-break repair in ferulic acid-treated animals [43]. The antioxidant ellagic acid, is a constituent but not a major component of BRBE; it exerts, at best, only a weak stimulatory effect on the DNA repair protein O^6^-methylguanine-DNA-methyltransferase in human lymphocytes and in glioblastoma and colon cancer cells [44]. Thus, it would be of interest to examine the role BRBE on the repair of the O^6^-methylguanine lesion since it is highly mutagenic in animals treated with environmental tobacco carcinogens [45].

In the present study, we showed for the first time that BRBE significantly enhances the NER efficiency observed with BP-dG and Gh lesions as the substrates. This enhancement is correlated with the upregulation of some critically important proteins that govern the NER process (Figure 9). Nevertheless, it is also important to recognize that there is a wide inter-individual variation in DNA repair capacities which could be due to polymorphisms in the genes encoding DNA repair proteins [46] However, BRBE had a much smaller impact on the BER pathway using Gh as the substrate. Despite its weak effect on BER activity, anthocyanins in BRB have been shown to scavenge reactive oxygen species (ROS) and thus protect cells from oxidative stress and DNA damage [47]. In a recent report, Pahlke et al., showed that anthocyanins lowered the basal level of ROS in human colon carcinoma cells [36]. We also showed that BRBE exerts antioxidant properties in human oral leukoplakia cells [22]. Therefore, in addition to its role as a powerful scavenger of ROS, using BRB as a whole food additive could be appealing for human consumption to enhance DNA repair capacity in high risk populations (smokers who are unable to quit, as well as non- or former-smokers who are exposed to environmental carcinogens) as a plausible agent for cancer prevention [48].

## Figures and Tables

**Figure 1 antioxidants-11-02110-f001:**
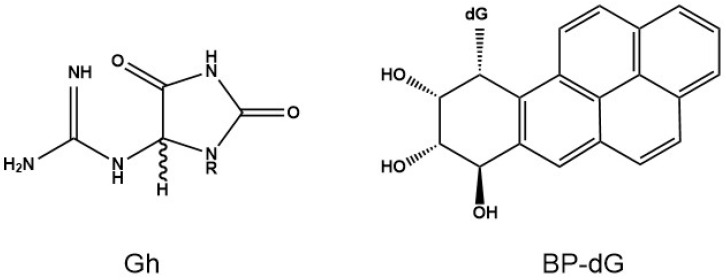
Structure of the guanidinohydantoin DNA lesion and benzo[*a*]pyrene diol epoxide-derived BP-dG adduct ((+)-*cis*-stereochemistry, covalently attached to the exocyclic amino group of guanine).

**Figure 2 antioxidants-11-02110-f002:**
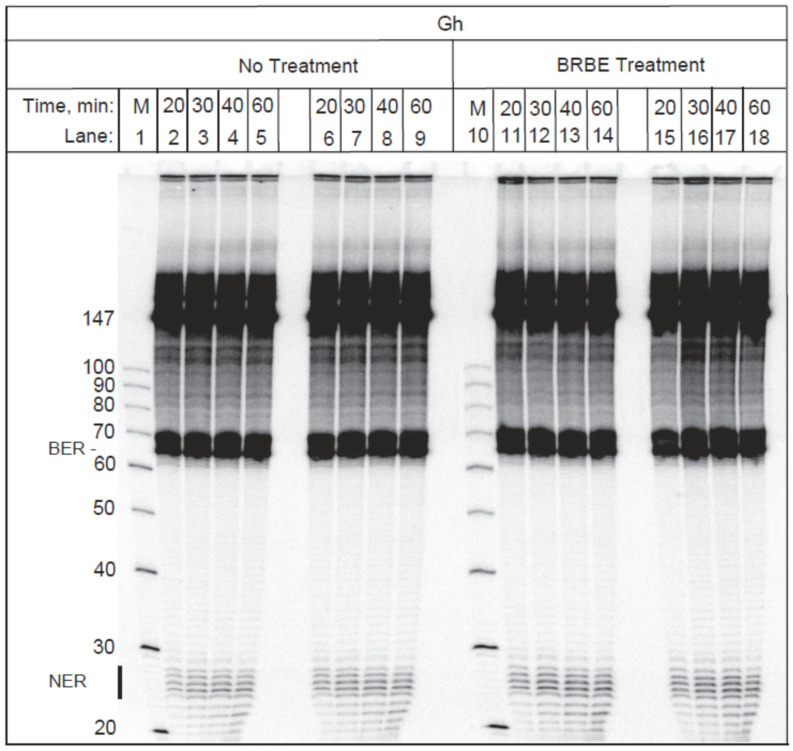
**Gh substrate.** Gel electrophoresis image of single guanidinohydantoin lesions (Gh) embedded in 147-mer DNA duplexes incubated in extracts from intact HeLa cells pre-treated or untreated with BRBE as a function of incubation time. The positions of the NER excision products are denoted by a vertical bar on the bottom left side of the gel.

**Figure 3 antioxidants-11-02110-f003:**
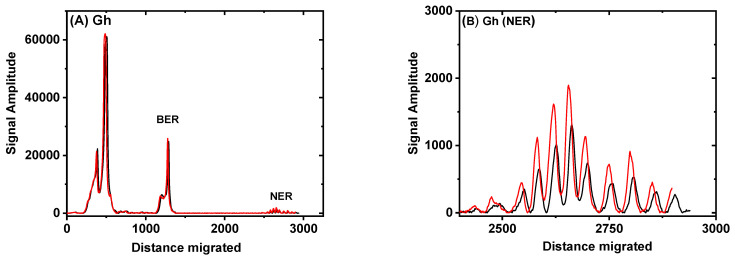
**Gh sample**, DNA repair product excision profiles: Gh lesion in 147-mer duplexes. Red lines: incubation in extracts from BRBE-treated HeLa cells; black lines: extracts from untreated cells. Densitometry scans of the 30 min incubation time lane in Figure 2. Panel (**A**): Trace of the entire lane, from the top to the bottom of the gel. Panel (**B**): Amplification (×20) of the lower region of the same lane showing the relative amplitudes of the ~24–30 nt oligonucleotide NER excision products.

**Figure 4 antioxidants-11-02110-f004:**
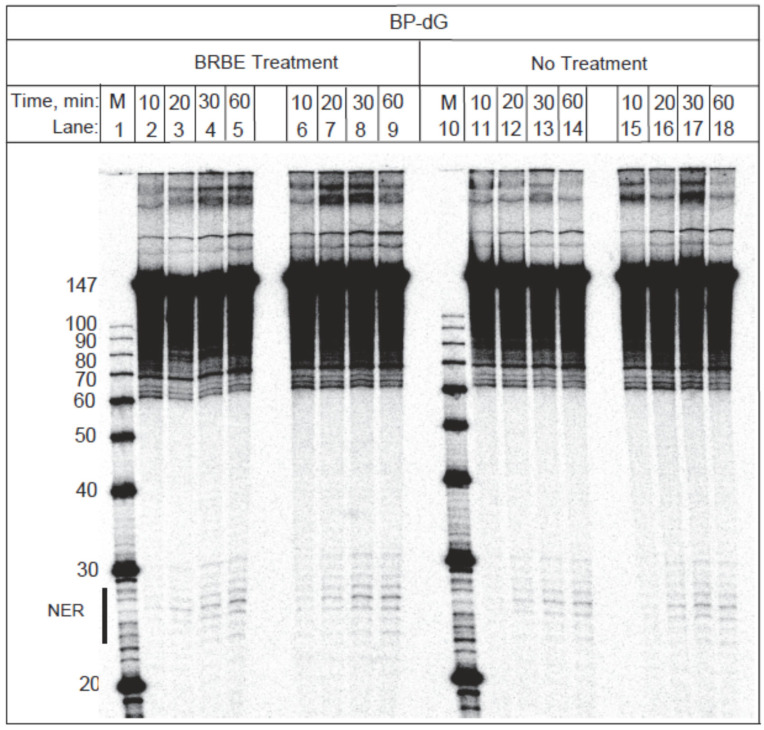
**BP-dG adduct substrates**. Gel electrophoresis image. Time dependence of formation of NER products of 147-mer DNA. Duplexes containing single benzo[a]pyrene diol epoxide-derived guanine adducts (BP-dG). After incubation in extracts from intact HeLa cells untreated or pre-treated with BRBE. The positions of the NER excision products are denoted by a vertical bar on the bottom left side of the gel. As expected, identical, but unmodified oligonucleotide sequences remain un-incised as demonstrated elsewhere [19,29], and which is consistent with the lXPD helicase-driven lesion verification mechanism of NER [30] that ensures that unmodified DNA sequences are not processed by the NER apparatus (see below).

**Figure 5 antioxidants-11-02110-f005:**
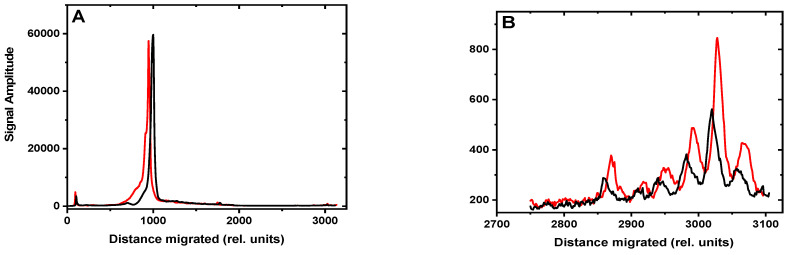
BP-dG adduct sample. Example of a densitometry scans of the 30 min incubation time lane derived from BRBE-pretreated (red trace) and untreated (black trace) HeLa cell extract NER product profile (data from Figure 4. Panel (**A**): Trace of the entire lane, from top to the bottom of the gel. Panel (**B**): ×100 amplification of the lower region of the lanes containing the ~24–30 nt oligonucleotide dual incision NER excision products (Figure 4).

**Figure 6 antioxidants-11-02110-f006:**
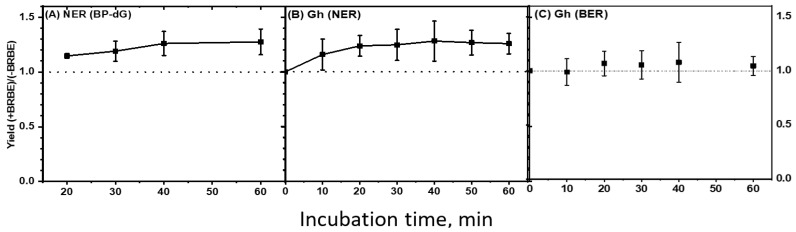
Time dependence of the ratios of NER product yields, [Y(+BRBE)/Y(−BRBE)] as a function of incubation time. (**A**). NER, BP-dG adduct. Each time point represents the average and standard deviations of 28 data points (see the text). The average value of all four time points shown is [Y(+BRBE)/Y(−BRBE]BP(NER) = 1.24 ± 0.09 (two-tailed *t*-test, *p*-value 0.05. (**B**). Gh lesion (NER). Time dependence of the ratios of NER product yields, [Y(+BRBE)/Y(−BRBE)], as a function of incubation time. Each time point represents the average of 32 data points, yielding an overall average value of the five time points shown [Y(+BRBE)/Y(−BRBE)]Gh(NER) = 1.24 ± 0.09 (two-tailed *t*-test, *p*-value 0.05. (**C**). Gh lesion (BER). Time dependence of the ratios of BER product yields, [Y(+BRBE)/Y(−BRBE)]Gh(BER) as a function of incubation time. Average value calculated as in panel B: [Y(+BRBE)/Y(−BRBE)]Gh(BER) = 1.06 ± 0.01 (*p*-value < 0.05).

**Figure 7 antioxidants-11-02110-f007:**
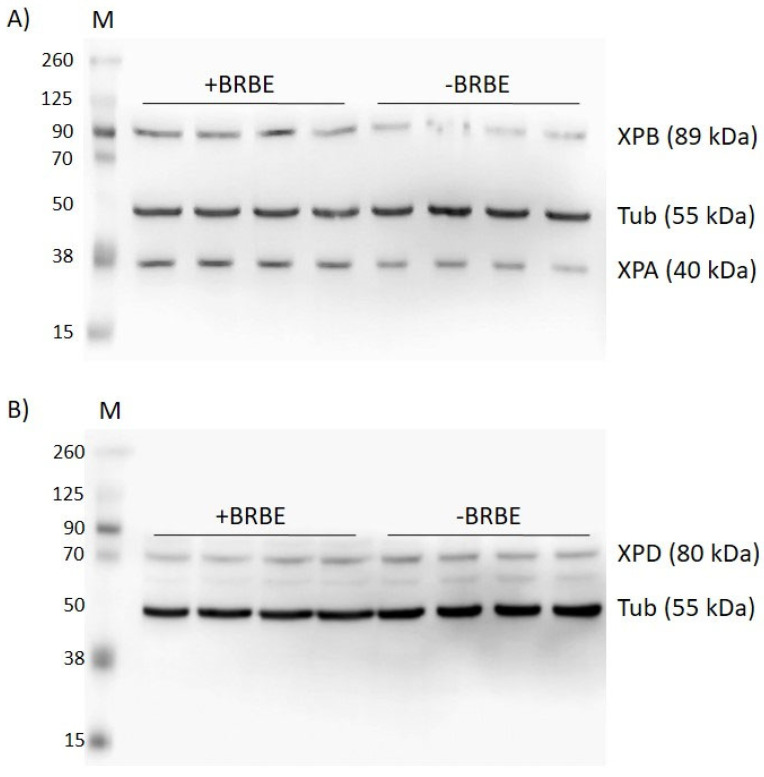
Enhancement of NER protein expression in BRBE-treated HeLa cells. (**A**) XPA, XPB and (**B**) XPD protein expression in HeLa cells, all with and without pre-treatment with BRBE. Tubulin is an inert cellular protein that was used as a control for the overall cellular protein concentrations in BRBE-treated and untreated cells. All Western Blot measurements of relative protein concentrations were conducted within the linear range of overall protein concentrations shown in Figure 8.

**Figure 8 antioxidants-11-02110-f008:**
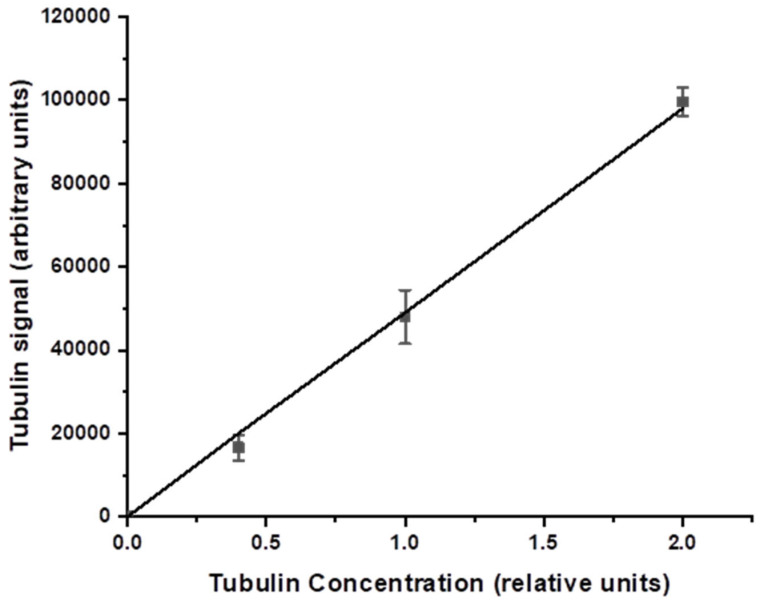
Western Blot signal as a function of tubulin concentration. The range of protein concentration for measuring the XPB, XPA and XPD levels in BRBE-treated and untreated HeLa cells were performed within the linear protein concentration range shown in this graph.

**Figure 9 antioxidants-11-02110-f009:**
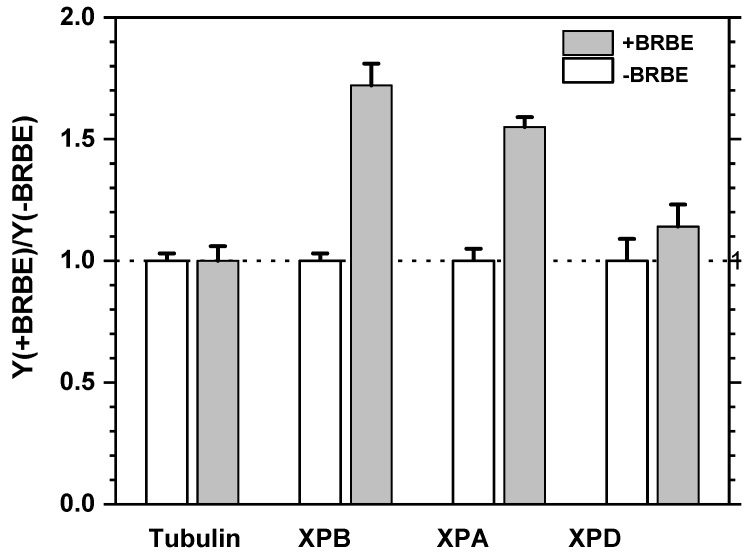
Effect of BRBE treatment of HeLa cells on the expression of selected NER proteins.

## Data Availability

The datasets generated and/or analyzed during the current study are available from the corresponding author upon request.

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
