# Peer review of "Treatment of Human HeLa Cells with Black Raspberry Extracts Enhances the Removal of DNA Lesions by the Nucleotide Excision Repair Mechanism"

_antioxidants, 2022, doi:10.3390/antiox11112110_

Round 1

Reviewer 1 Report

This manuscript investigates the potential effect of black raspberry extracts (BRBE) on the DNA repair capacity of human cells. The authors tested cellular repair capacity using whole-cell extracts and oligonucleotides bearing DNA damage at given positions. In one case it is a carcinogen-induced bulky adduct that is subject to nucleotide excision repair (NER) and in the other case, it is an oxidative stress-induced non-bulky DNA lesion that is repaired through both NER and base excision repair (BER). The distinct patterns of incisions of the damage-containing oligonucleotides due to the activity of BER or NER proteins were followed. The authors observed significantly increased activity of NER after BRBE treatment, whereas the change in BER activity was insignificant. The increased NER activity corresponded well with an increased cellular expression of selected NER factors, as demonstrated by western blot experiments. In general, the experiments seem to be correct, with controls. However, an important control is missing from Figures 2 and 4, which should show the incision reaction products of an oligonucleotide corresponding to the ones used in the experiments, but without any DNA damage.

Overall, this manuscript makes interesting observations and establishes a link between BRBE treatment and increased NER activity. It has been previously shown that BRB inhibits carcinogenesis in animal models and DNA damage but its stimulating effect on DNA repair is a novel finding.

Minor point: In figures 2 and 4 there are no indications why the time points are duplicated both with and without treatment. Are these replicates?

Author Response

Reviewer 1

This manuscript investigates the potential effect of black raspberry extracts (BRBE) on the DNA repair capacity of human cells. The authors tested cellular repair capacity using whole-cell extracts and oligonucleotides bearing DNA damage at given positions. In one case it is a carcinogen-induced bulky adduct that is subject to nucleotide excision repair (NER) and in the other case, it is an oxidative stress-induced non-bulky DNA lesion that is repaired through both NER and base excision repair (BER). The distinct patterns of incisions of the damage-containing oligonucleotides due to the activity of BER or NER proteins were followed. The authors observed significantly increased activity of NER after BRBE treatment, whereas the change in BER activity was insignificant. The increased NER activity corresponded well with an increased cellular expression of selected NER factors, as demonstrated by western blot experiments. In general, the experiments seem to be correct, with controls. However, an important control is missing from Figures 2 and 4, which should show the incision reaction products of an oligonucleotide corresponding to the ones used in the experiments, but without any DNA damage.

Lack of unmodified DNA controls. We cite previous results that clearly show experimentally that DNA duplexes without DNA damage do not yield any of the characteristic excision NER product yields 24 – 32 nt in length. These products arise only because the helicase XPD verifies the presence of a DNA lesion that stalls the progress of this helicase which is the signal for the recruitment of the endonucleases XPD and XPF to TFIIH that incise the damaged DNA duplex. Without stalling of the helicases, the NER process is cancelled.  These reminders are stated in lanes 204-207 in the Figure 2 legend with two previously published references that experimentally demonstrate and confirm these facts using similar oligonucleotides. We supply a short discussion of the NER mechanism in lines

 (see We also added the following text to the legend to Figure 2 that summarizes the basics of the complicated NER mechanism:

“Identical, but unmodified oligonucleotide sequences remain un-incised as demonstrated elsewhere [19, 29], and which is consistent with the XPD helicase-driven lesion verification mechanism of NER[30].  This mechanism ensures that unmodified DNA sequences are not processed by the NER apparatus below).”

Overall, this manuscript makes interesting observations and establishes a link between BRBE treatment and increased NER activity. It has been previously shown that BRB inhibits carcinogenesis in animal models and DNA damage but its stimulating effect on DNA repair is a novel finding.

Minor point: In figures 2 and 4 there are no indications why the time points are duplicated both with and without treatment. Are these replicates?

Yes, Lanes 1-5 and 6-9  are replicates to verify  reproducibility in each set of experiments.

Reviewer 2 Report

Sales and colleagues have explored the effects of blackberry extracts on the DNA repair activity of HeLa cells. In particular, they have assessed the levels of base excision (BER) and nucleotide excision (NER) repair activity and the abundance of selected NER factors in cells treated or not with such extracts for 24h. This work builds on several earlier studies by the same authors on the protective effects of blackberry extracts. Overall, the work has been well performed and the results are clearly presented. I suggest below a few modifications and additional experiments that could strengthen this study.

1.     In the timecourse experiments (Fig. 2 and 4) there seem to be little change between 20 and 60 minutes. Why did the authors not evaluate the repair activity at earlier timepoints as in the JBC2016 paper? Earlier time points may reveal more differences.

2.     For Fig. 2 and Fig. 4 gels, it would be good to provide gel images with reduced exposure to better assess BER activity. Also, a close-up comparison of BER activity, as was done in Fig.3 for NER, would be interesting.

3.     In the section “Effects of BRBE pretreatment of human cells on the NER of bulky BP-dG adducts”, the authors don’t describe their results presented in Fig. 4 and S1.

4.     Performing similar experiments on an 8-oxo-G substrate (only repaired by the BER) would be very interesting and could potentially strengthen the authors conclusions if no effects are seen.

5.     In the discussion, authors should clearly state that the BRB extracts increase overall NER activity by increasing the abundance of NER factors and not by modulating the repair activity directly. This is not always clear as it is stated presently (eg. line 380).

6.     XPA, XPB and XPD are relevant factors, but following the levels of XPC-RAD23B would certainly also be informative. It would also be interesting to see whether this increased levels comes from increased transcription (compare RNA levels) or increased translation and/or stability of the proteins.

7.     Minor comments: Line 228: replace “Since it is known” by “As expected since”. Is Fig. S1 a supplementary figure? What would be the effect of a longer pre-treatment with BRB extracts?

Author Response

Reviewer 2

Sales and colleagues have explored the effects of blackberry extracts on the DNA repair activity of HeLa cells. In particular, they have assessed the levels of base excision (BER) and nucleotide excision (NER) repair activity and the abundance of selected NER factors in cells treated or not with such extracts for 24h. This work builds on several earlier studies by the same authors on the protective effects of blackberry extracts. Overall, the work has been well performed and the results are clearly presented. I suggest below a few modifications and additional experiments that could strengthen this study.

Authors comment: The document has been revised and the reviewer’s comments and questions have been fully addressed. The spelling has been checked using the MS WORD program.

  1. In the time course experiments (Fig. 2 and 4) there seem to be little change between 20 and 60 minutes. Why did the authors not evaluate the repair activity at earlier timepoints as in the JBC2016 paper? Earlier time points may reveal more differences.

Yes, this is an excellent question. In fact, our measurements included  the 10 min time point (Figure 5B), as in the 2016 paper. However, the NER product yields are much lower than the 20 min time point, and the error bars are too large for assessing product yield ratios in the ~ 10 min time interval. These observations discouraged more detailed investigations below the 20 min time point.  

Figures S1 and S2 have now been merged with the text and renamed as Figures 3 and 9, respectively. The objective was to improve the clarity of the experimental gel results.

  1. For Fig. 2 and Fig. 4 gels, it would be good to provide gel images with reduced exposure to better assess BER activity. Also, a close-up comparison of BER activity, as was done in Fig.3 for NER, would be interesting.

  1. In the section “Effects of BRBE pretreatment of human cells on the NER of bulky BP-dG adducts,” the authors don’t describe their results presented in Fig. 4 and S1.

This concern (no discussion of former Figs. 4 (Densitometry scan of BP-dG gel); S1 (Tubulin calibration), are now Figures 3 and 9, respectively. This concern is now addressed in the following sections of the manuscript (lines 232-258 and 269-276).

  1. Performing similar experiments on an 8-oxo-G substrate (only repaired by the BER) would be very interesting and could potentially strengthen the authors conclusions if no effects are seen.

This is a good point. Yes, it would be a good supporting result. However, on the basis of the lack of effect of BRB on the BER of Gh, similar results with 8-oxodG would be expected, and we instead focused on the Western Blot experiments.  

  1. In the discussion, authors should clearly state that the BRB extracts increase overall NER activity by increasing the abundance of NER factors and not by modulating the repair activity directly. This is not always clear as it is stated presently (eg. line 380).

             Agreed, we added the following sentences:

Based on these findings, we conclude that enhanced expression of the NER factors XPB and XPD may play a role in the enhancement of NER activities in BRBE-treated cells. Thus, BRB extracts increase the overall NER activity by increasing the abundance of NER factors, and not by directly modulating the repair activity on lines 424-427.

  1. XPA, XPB and XPD are relevant factors, but following the levels of XPC-RAD23B would certainly also be informative. It would also be interesting to see whether this increased level comes from increased transcription (compare RNA levels) or increased translation and/or stability of the proteins.

Yes, these are interesting leads that will need to be investigated in the future studies.

  1. Minor comments: Line 228: replace “Since it is known” by “As expected since”.

Done!

Is Fig. S1 a supplementary figure? Yes! The two supplementary Figures have now been inserted into the main text to enhance the clarity of the discussion of the data.

  What would be the effect of a longer pre-treatment with BRB extracts?

We selected the 24 h time point for the following reason: Most of the BRB components are water-soluble. and would be expected to be excreted within 24 h in vivo. We therefore anticipated that the effects of BRB beyond ~ 24 h would be declining.

Round 2

Reviewer 1 Report

The authors have addressed my concerns.